# Multi-View Radiomics Feature Fusion Reveals Distinct Immuno-Oncological Characteristics and Clinical Prognoses in Hepatocellular Carcinoma

**DOI:** 10.3390/cancers15082338

**Published:** 2023-04-17

**Authors:** Yu Gu, Hao Huang, Qi Tong, Meng Cao, Wenlong Ming, Rongxin Zhang, Wenyong Zhu, Yuqi Wang, Xiao Sun

**Affiliations:** 1State Key Laboratory of Bioelectronics, School of Biological Science and Medical Engineering, Southeast University, Nanjing 210096, China; guyu_bio@seu.edu.cn (Y.G.); haohuang@seu.edu.cn (H.H.);; 2Department of Radiology, Nanjing Drum Tower Hospital, The Affiliated Hospital of Nanjing University Medical School, Nanjing 210008, China; 3Department of General Surgery, Nanjing Drum Tower Hospital, The Affiliated Hospital of Nanjing University Medical School, Nanjing 210008, China

**Keywords:** radiogenomics, radiomics, hepatocellular carcinoma, contrast-enhanced CT, tumor heterogeneity, prognosis, data fusion

## Abstract

**Simple Summary:**

Hepatocellular carcinoma is a widespread cancer with complex molecular heterogeneity. Compared with invasive tissue sampling, the radiomics framework shows promise in non-invasively decoding tumor heterogeneity. In this study, we utilized integrative analysis of radiomics and genomics profiles to characterize hepatocellular carcinoma inter-tumor and intra-tumor heterogeneity. We extracted multi-view imaging features from contrast-enhanced CT scans, and fused features for potential radiomics subtypes identification. Differentiated immune pathway activity and inflammatory tumor microenvironment between subtypes were obtained, and the predominant radiogenomics association between texture-related and immune-related was demonstrated and validated in independent cohorts. These findings could provide clues for non-invasive inflammation-based risk stratification in hepatocellular carcinoma.

**Abstract:**

Hepatocellular carcinoma (HCC) is one of the most prevalent malignancies worldwide, and the pronounced intra- and inter-tumor heterogeneity restricts clinical benefits. Dissecting molecular heterogeneity in HCC is commonly explored by endoscopic biopsy or surgical forceps, but invasive tissue sampling and possible complications limit the broadeer adoption. The radiomics framework is a promising non-invasive strategy for tumor heterogeneity decoding, and the linkage between radiomics and immuno-oncological characteristics is worth further in-depth study. In this study, we extracted multi-view imaging features from contrast-enhanced CT (CE-CT) scans of HCC patients, followed by developing a fused imaging feature subtyping (FIFS) model to identify two distinct radiomics subtypes. We observed two subtypes of patients with distinct texture-dominated radiomics profiles and prognostic outcomes, and the radiomics subtype identified by FIFS model was an independent prognostic factor. The heterogeneity was mainly attributed to inflammatory pathway activity and the tumor immune microenvironment. The predominant radiogenomics association was identified between texture-related features and immune-related pathways by integrating network analysis, and was validated in two independent cohorts. Collectively, this work described the close connections between multi-view radiomics features and immuno-oncological characteristics in HCC, and our integrative radiogenomics analysis strategy may provide clues to non-invasive inflammation-based risk stratification.

## 1. Introduction

Hepatocellular carcinoma (HCC), also known as malignant hepatoma, is the third-leading cause of cancer death worldwide [1]. Risk factors for HCC include chronic hepatitis B/C infection, cirrhosis linked to alcohol abuse, diabetes, and obesity [2]. Unfortunately, only 30% of cases could be diagnosed at early stages, with a 5-year recurrence rate as high as 50% to 60% [3,4]. More patients diagnosed at advanced stages have limited clinical benefit in chemotherapy and radiotherapy. The development of HCC is a multifaceted process that involves an intricate interplay between altered signaling pathways, tumor microenvironment, and diverse genetic profiles, which leads to high tumoral heterogeneity, and ultimately poses a formidable obstacle to deciphering therapeutic strategies [5]. In general, tumor heterogeneity can be categorized as inter-tumor (tumor by tumor) and intra-tumor (within a tumor). Inter-tumor heterogeneity arises from altered phenotypes induced by various environmental and etiological factors among patients [6], and intra-tumor heterogeneity refers to the diversification of both malignant and nonmalignant components (endothelial, stromal, and immune cells, among others) of the tumor microenvironment within a single tumor lesion [7]. Given the complexity and heterogeneity, characterizing the landscape of HCC is paramount. Recently, heterogeneity exploration based on multi-omics molecular profiles, such as gene expression, methylation, and immunogenomics, has been conducted [8,9] and brings insights for prognosis prediction and immune response evaluation. However, invasive tissue sampling and possible complications limit the broader adoption. Thus, the development of non-invasive diagnostic and prognostic methods or markers is highly needed.

Imaging examination is a common tool for diagnosis, staging, treatment guidance, and response monitoring in HCC [10]. In the era of personalized oncology, radiomics adds layers of complexity and resolution to conventional imaging, which converted standard radiographic medical images into a high-dimensional quantitative feature space, providing valuable information on tumor pathophysiology and molecular subtyping [11,12,13]. A previous systematic review of 54 included HCC studies has acknowledged the predictive value of radiomics features [14]. For example, Asayama et al., Defour et al., and Xu et al. demonstrated the good predictive performance of radiomics features in diagnosis [15], prognosis [16], and microvascular invasion [17], respectively. For pathologic and molecular correlation, Chen et al. uncovered features including the peritumoral region associated with tumor-infiltrating lymphocyte population [18]. For treatment response, Kim et al. revealed the predictive contribution of radiomic features in post-TACE overall survival [19]. Recently, the integrative analysis of imaging phenotypes with genomic-level information, called radiogenomics [20], has gained increasing attention for revealing the relevance to cancer development and progression. For instance, histopathological features and genomics features have shown close associations with imaging features in several HCC studies [21,22,23]. While these initial reports are promising, most of them are only focused on clinical features and typical functional gene expression programs, and rarely dissect the intra-tumor heterogeneity of immunobiology in the tumor immune microenvironment (TIME). The information shared in phases and regions could also be overlooked by simply utilizing single-phase images or coarsely merging multi-view imaging features. Therefore, the biological significance of the radiomics signatures and the value of multi-view imaging features for noninvasive prediction of immuno-oncologic characteristics and prognoses require further investigation.

In this study, we aimed at dissecting inter-tumor and intra-tumor heterogeneity of HCC based on integrative analysis of radiomics and genomics profiles. By utilizing a multi-view fused imaging feature subtyping model, we identified radiomics subtypes with distinct biological significance and inflammatory TIME status. Our study highlights the radiogenomics linkages between multi-view CE-CT imaging features and immuno-oncological characteristics, and could provide the theoretical rationale and feasibility for non-invasive inflammation-based risk stratification.

## 2. Materials and Methods

### 2.1. Data Collection

The CE-CT images of 40 HCC patients were collected from The Cancer Imaging Archive (TCIA, https://www.cancerimagingarchive.net/, accessed on 1 April 2021) [24]. By excluding eight patients with missing images in the enhanced phase or with existing metal artifacts and two patients without reference tumor location, a total of 30 patients with complete imaging data and clinical information were eventually included for downstream analysis. The corresponding transcriptomics data, exacted from The Cancer Genome Atlas Liver Hepatocellular Carcinoma (TCGA-LIHC) cohort, were downloaded from the GDC data portal (https://portal.gdc.cancer.gov, accessed on 1 July 2021). Besides, 192 HCC patients without images in TCGA-LIHC (https://portal.gdc.cancer.gov/projects/TCGA-LIHC, accessed on 1 July 2021) and 142 HCC patients in Liver Cancer-NCC, JP (LINC-JP, https://dcc.icgc.org/releases/current/Projects/LINC-JP#!, accessed on 1 July 2021) [25] were obtained as two additional cohorts for further radiogenomics association validation analysis. Table 1 shows detailed baseline characteristics, clinical data, and follow-up information of all patients involved in our study.

### 2.2. Volume of Interest Segmentation and Radiomics Feature Extraction

The multi-view VOIs consisted of tumor and peritumoral regions in the arterial and venous phases. The tumor reference coordinates were obtained from the crowd-sourcing website [26], which generated consensus tumor markups by radiologists partnered with the Radiological Society of North America. The multi-view VOIs were divided into four types: (1) tumor region in the arterial phase, (2) tumor region in the portal venous phase, (3) peritumoral region in the arterial phase, and (4) peritumoral region in the portal venous phase (Figure 1 and Appendix A). For tumoral VOIs, the lesion was manually annotated on both arterial and portal venous phase images based on reference markups, and segmented using a threshold-based segmentation algorithm by 3d slicer software (version 4.9.0, http://www.slicer.org, accessed on 1 April 2021). For peritumoral VOIs, we performed a morphologic dilation operation to capture the peritumoral region of the entire tumor VOIs using the open-source library OpenCV (version 4.4.0), with a radial distance of 10 mm based on a previous study [27]. The area beyond the liver parenchyma was removed, and large vessels, adjacent organs, and air cavities were also excluded.

Image preprocessing and feature extraction were performed using the Pyradiomics tool (version 3.0.1) in Python 3.7 [28]. Images were resampled to a voxel size of 1 × 1 × 1 mm to standardize the voxel spacing and discretized with a fixed bin width of 25. The radiomics features in each VOI were exacted and comprised of three classes: (1) first-order features (*n* = 18), (2) shape features (*n* = 14), and (3) texture features (*n* = 68). First-order features describe the distribution of voxel intensities, and shape features describe the difference in the shape of tumors. Texture features describe the gray profile and were further subdivided into the gray level co-occurrence matrix (GLCM), gray level dependence matrix (GLDM), gray level run length matrix (GLRLM), and gray level size zone matrix (GLSZM). Finally, we extracted multi-view radiomics features for each lesion. A full list of the features is described in Appendix A, and eigenvalues were standardized with Z-score for further analysis.

### 2.3. Radiomics Feature Fusion and Subtype Identification

Owing to multi-view imaging features reflecting tumor heterogeneity from various regions and phases, it is crucial to choose an appropriate method to fully use the knowledge provided by multi-view features, instead of simple concatenation. Here, we developed the fused imaging feature subtyping (FIFS) model to better distinguish intrinsic subtypes in HCC patients. The FIFS model was built using the similarity network fusion (SNF) algorithm in R package SNFtool (version 2.3.1) [29]. Specifically, we first constructed the patient-by-patient similarity network for each type of multi-view radiomics feature, respectively, obtaining a total of four networks. In these networks, the nodes represented patients and the weighted edges represented patient–patient similarities. We then employed a nonlinear network fusion method, called message-passing theory, to iteratively integrate these networks. After iteratively updating, networks converged to a fusion network, with low-weight edges in networks discarded and high-weight edges retained in the fusion network. Finally, we used this fusion network to cluster the patients into different radiomics subtypes.

To determine the optimal number of subtypes in the final clustering step, we applied the consensus clustering method using R package ConsensusClusterPlus (version 1.58.0) [30], which provides a quantitative assessment for determining the number of possible clusters. We tested different cluster groups and performed 1000 iterations with resampling of 80%. The optimal clustering number was determined according to consensus matrices and cumulative distribution functions (CDFs), which are commonly used to optimize clustering stability. Additionally, we calculated the silhouette score to measure the homogeneity of the subtypes and calculated the log-rank test P to assess the prognostic quality [29]. A higher silhouette score indicates that the patients are well matched to their own cluster and poorly matched to neighboring clusters. The lower the log-rank test P, the more significant the prognostic effect observed. Consequently, based on multi-view CE-CT images, we applied the FIFS model to identify two distinct radiomics subtypes, named FIFS1 and FIFS2.

### 2.4. Functional Enrichment Analysis and TIME Comparison between FIFS Subtypes

By integrative analysis of radiomics and genomics profiles, we dissected tumor heterogeneity from the perspective of intra-tumor and inter-tumor, respectively. To evaluate the heterogeneity of biological pathways functions and TIME between two subtypes, we employed the transcriptomics data corresponding to the patients. DEGs were identified using R package DESeq2 [31] (version 1.18.1) by comparing each pair of subtypes. The filtering threshold for the DEGs was set as follows: log2 fold change > 1 or log2 fold change < −1, with adjusted *p* < 0.05. The DEGs were used for functional enrichment analysis based on the KEGG [32] and GO [33] databases using R package clusterProfiler (version 3.8.1) [34]. Hallmark pathways were collected from the MSigDB database [35] and used for GSVA by R package GSVA (version 1.40.1) [36].

The immune and stromal infiltration scores of tumor microenvironments were calculated using R package ESTIMATE (version 1.0.13) [37]. The tumor purity was inferred by multiple algorithms based on previous research, including the ESTIMATE algorithm, the LUMP algorithm, immunohistochemistry (IHC) qualitative estimation, and the consensus measurement of purity estimation (CPE) algorithm [38]. The abundance of individual immune cell types in TIME was inferred using the CIBERSORT interface [39], which estimated the relative fraction of immune cell types based on the deconvolution method. The gene sets of immune-related molecules were compared by GSVA, including cell surface immune-related molecules (costimulators, coinhibitors, and major histocompatibility complex) and cytokines (interleukins, chemokines, interferons, and colony-stimulating factors) [40]. Tumor immunogenicity was calculated by immunophenoscore (IPS) based on the gene expression in effector cells, immunosuppressive cells, MHC molecules, and immunomodulators [41]. Stemness indices (mRNAsi score) were calculated based on the Malta TM method and the one-class logistic regression (OCLR) machine-learning algorithm program [42].

### 2.5. Comparative Analysis with Previous Molecular Subtypes

To investigate the potential association between the subtypes identified by the FIFS model and existing molecular subtyping systems, we compared our FIFS results to molecular subtyping results derived from three prior studies. The first study was performed by Hoshida et al. [43], which incorporated nine independent cohorts to identify three distinct molecular subtypes correlated with clinical features (termed S1, S2, and S3). S1 was characterized by WNT pathway activation. S2 was characterized by poor differentiation, a high level of AFP expression, and enrichment in MYC pathway and AKT pathway activation. S3 was characterized by good differentiation and molecular program of differentiated hepatocyte function. In the second study, TCGA research network integrated multi-omics profiles to identify three subtypes of HCC (termed iCluster1–iCluster3) [8]. iCluster1 was characterized by an immune low signature, iCluster 2 was summarized as having a non-proliferative status, and iCluster3 was classified as immune exhausted. PanImmune was established in the third study to identify six immune subtypes, termed wound healing (C1), IFN-gamma dominant (C2), inflammatory (C3), lymphocyte depleted (C4), immunologically quiet (C5), and TGF-beta dominant (C6) [9]. The significance of the overlap between our radiomics subtypes and other subtype systems was assessed using the hypergeometric test.

### 2.6. Radiogenomics Association Identification and Validation

For evaluation of the radiogenomics association in our FIFS model, multiple steps were performed as follows: (1) WGCNA was employed to identify the subtype-specific co-expression modules based on R package WGCNA (version 1.71) [44]. In detail, a signed weighted correlation matrix, which contained pairwise Pearson correlations between all genes across patients, was generated using a soft threshold of β = 8 to reach a scale-free topology. The dynamic hybrid tree-cut method was used to detect the network modules of co-expressed genes with a minimum module size of 30. The FIFS subtyping results were converted into a 0–1 phenotypic matrix to identify subtype-specific modules, (2) the PRF-related modules were identified by filtering based on Cox regression analysis and correlation analysis between the modules and radiomics features, (3) the biological function of the PRF-related modules was assessed by GO and KEGG enrichment analysis, and (4) we constructed pathway–feature pairs between the PRF-related modules and imaging features based on the GSVA method and correlation analysis and identified the specific pathway and gene for each feature. Cytoscape (version 3.9.1) was employed to visualize the association network of imaging features and biological functions.

For validation of the radiogenomics association, a gene-expression-based classifier was constructed to predict the potential FIFS subtype on two additional cohorts using the nearest shrunken centroid method by R package pamr (version 1.56.1) [45]. Specifically, we applied two steps to obtain the most representative signature genes using 30 patients with the FIFS subtyping information as the training set (16 FIFS1 and 14 FIFS2 patients). First, the overlapped genes of DEGs between FIFS subtypes and the genes in PRF-related modules were identified. Second, through 10-fold cross-validation with stratified sampling, we filtered the overlapped genes based on the smallest preselected gene number and the lowest misclassification rate. The retained genes were used to build the classifier. We applied this classifier to two additional cohorts to classify patients into the FIFS subtype. Using Kaplan–Meier survival analysis [46], overall survival was compared between patients. The innate and adaptive immune-related pathways were collected from a previous study [47] and immunocompetent status was evaluated using GSVA.

### 2.7. Quantification and Statistical Analysis

Continuous variables were compared using the Student’s *t*-test or Wilcoxon test, and the Shapiro–Wilk test was used to test the normality of distributions. Categorical variables were compared using Pearson’s chi-square test or Fisher’s exact test. Survival analysis was conducted using the Kaplan–Meier method and compared with the log-rank test. Regression analysis was performed to ascertain prognostic radiomics features using a Cox proportional hazards model. All statistical tests were two-sided and the statistical significance threshold was set at *p* < 0.05. The false discovery rate (FDR) correction was used in multiple hypothesis testing to decrease false positive rates. Statistical analyses were performed with R software (version 4.1.1, https://www.R-project.org/, accessed on 1 January 2022).

## 3. Results

### 3.1. Study Design

The overall experimental design is depicted in Figure 1, including three steps: (1) radiomics feature extraction and fused imaging feature subtyping (FIFS) model construction, (2) radiomics subtype identification and tumor heterogeneity comparison, and (3) radiogenomics analysis (Figure 1). Specifically, based on CE-CT imaging of HCC patients, we annotated and segmented the multi-view volumes of interest (VOIs). Then, we extracted the features for each VOI and developed the FIFS model to fuse multi-view features followed by radiomics subtype identification. Based on corresponding transcriptomics data, we compared inter-tumor and intra-tumor heterogeneity between subtypes, and finally assessed the radiogenomics association and validated our findings using independent cohorts.

### 3.2. Identifying HCC Imaging Subtypes Based on Multi-View Radiomics Feature Fusion

To dissect tumor heterogeneity based on radiomics in HCC, we first extracted multi-view CE-CT imaging features of HCC patients collected from The Cancer Imaging Archive (TCIA) database. After imaging quality screening, 30 HCC samples were included for downstream analysis (Figure 2A). Multi-view VOIs were annotated and segmented based on reference tumor location, including (1) type 1, the tumor region in the arterial phase; (2) type 2, the tumor region in the portal venous phase; (3) type 3, the peritumor region in the arterial phase; and (4) type 4, the peritumor region in the portal venous phase, respectively (Figure 1 and Figure 2A). For each type of VOI, a total of 100 radiomics features comprised of three classes were extracted: (1) first-order features (*n* = 18), (2) shape features (*n* = 14), and (3) texture features (*n* = 68). A full list of the features is described in Appendix A.

Instead of coarsely merging features, we developed the FIFS model to identify potential radiomics subtypes. The FIFS model utilized all four types of multi-view radiomics features (defined as combination pattern, CP1, Appendix A) as input and learned a matrix of similarities between features in each type by network fusion. After determining the optimal number of clusters based on consensus matrices and cumulative distribution functions (CDF, Appendix A), we identified two distinct radiomics subtypes, FIFS1 and FIFS2 (Figure 2B). Principal component analysis of HCC samples based on fused features could distinguish the two subtypes well (Figure 2C). To further evaluate whether our radiomics subtyping results were prognostically relevant, we performed survival analysis and found that lower overall survival rates (*p* = 0.004, log-rank test) and lower disease-free survival rates (*p* = 0.039, log-rank test) were observed in FIFS2 compared with FIFS1 (Figure 2D,E).

To examine whether all four types of multi-view features contribute to the FIFS model, we compared three other alternative feature inputs, including (1) CP2, the features of tumor in the arterial and the venous phase (type 1 and type 2); (2) CP3, the features of tumor and peritumor in the venous phase (type 2 and type 4); and (3) CP4, the features of tumor and peritumor in the arterial phase (type 1 and type 3) (Appendix A). The consensus matrices and CDF were also applied to optimize the clustering stability, and the silhouette score and overall survival rate were used to evaluate the clustering effects (Appendix A). As a result, two clusters were determined as the optimal cluster number for each CP (Appendix A). In comparison with another three CPs, CP1 showed the highest silhouette score and the most significant prognostic stratification (Appendix A) and demonstrated that integrating CP1 based on the FIFS model was a desirable strategy. Altogether, we extracted and integrated multi-view features by the FIFS model and identified two robust radiomics subtypes with prognostic relevance for subsequent analysis.

### 3.3. Radiomics Subtypes Describe Distinct Texture-Dominated Imaging Profiles

We compared the clinical features between FIFS1 and FIFS2 to dissect the clinical relevance. Consequently, a high histologic tumor grade (G3) was found to be more common in FIFS2, while FIFS1 displayed a lower histologic grade (G1, *p* < 0.001, Fisher’s exact test, Figure 3A and Appendix A). Moreover, serum alpha-fetoprotein (AFP) values were significantly higher in FIFS2 than in FIFS1 (*p* = 1.1 × 10^−3^, two-sample Student’s *t*-test, Figure 3B). Furthermore, we found that the radiomics subtype identified by the FIFS model was an independent prognostic factor (Figure 3C, *p* = 0.014, HR = 0.146, Cox regression analysis).

As previous studies have demonstrated the clinical and prognostic relevance of imaging features, we also evaluate the prognostic significance of the subtype-specific radiomics features. Based on eigenvalues of radiomics features, 142 differential features were obtained between two subtypes (*p* < 0.05, two-sample Student’s *t*-test), and 18 of them were related to prognosis (*p* < 0.05, Cox regression analysis), named prognostic radiomics features (PRFs) 1–18 (Table 2, Figure 3D, Appendix A). By observing the distribution of PRFs in four VOI types, more than half fell into type 2 (Table 2). Furthermore, the PRFs were mainly distributed in the venous phase (14/18) and tumor region (13/18), and most of them (15/18) belonged to the texture class (Table 2). Compared with FIFS2, we found FIFS1 had significantly higher eigenvalues in PRF1–10, which was characterized by a higher low gray value (PRF3, 4, 6, 7, 8) and lower local homogeneity (PRF12, 13), as well as a greater surface–volume ratio (PRF9, S/V ratio) in the tumor region (Figure 3D). By contrast, FIFS2 had significantly higher eigenvalues in PRF11–18, which was characterized by a stronger high gray value (PRF14, 15) and higher local homogeneity (PRF12, 13) in the tumor region (Figure 3D). In addition, the heatmap showed that the weaker heterogeneity (PRF17, 18) and high gray value (PRF2, 5) profiles in the peritumoral region were observed in FIFS1, and stronger heterogeneity (PRF17, 18) and low gray value profiles (PRF16) in the peritumoral region were presented in FIFS2 (Figure 3D, Table 2). These results demonstrated the distinct texture-dominated imaging profiles between FIFS1 and FIFS2.

To investigate whether the FIFS model may reflect inter-tumoral biological characteristics in HCC, we compared the subtyping results based on FIFS and the three molecular subtyping systems previously established (Figure 3E). Consistent with our findings in clinical comparison, we found that FIFS2 had a greater proportion of Class S2 tumors in Hoshida subtyping [43], characterized by high levels of AFP expression and enrichment in MYC pathway (*p* = 0.034, hypergeometric test). Moreover, in iCluster subtyping [8], FIFS1 had a higher proportion of iCluster2, characterized by non-proliferative, lower tumor grade, and better prognosis (*p* = 0.015, hypergeometric test). In PanImmune subtyping [9], the comparison revealed that FIFS1 had a higher proportion of C2 (IFN-gamma dominant), showing strong CD8 signals and the highest M1/M2 macrophage polarization and the greatest TCR diversity (*p* = 0.013, hypergeometric test). Meanwhile, FIFS2 had a higher proportion of C4 (lymphocyte depleted), showing more M2 macrophage polarization infiltrate and worse prognosis (*p* = 0.001, hypergeometric test). In summary, we uncovered the difference in clinical characteristics and radiomics patterns between FIFS1 and FIFS2, and demonstrated the association with the established molecular subtyping systems.

### 3.4. Distinct Biological Significance and Proinflammatory TIME Status of the FIFS Subtypes

To dissect the heterogeneity of biological function and TIME, we performed a comparative transcriptomics analysis of the FIFS subtypes. We identified 1966 differentially expressed genes (DEGs) between FIFS1 and FIFS2 samples (Figure 4A, Appendix A). Gene Ontology (GO) and Kyoto Encyclopedia of Genes and Genomes (KEGG) analysis showed distinct biological function enrichment (*p* < 0.05, hypergeometric tests, Figure 4B and Appendix A). More specifically, FIFS1 was significantly enriched in proinflammatory functions, including cytokine activity, chemokine signaling, and interleukin and interferon production. Meanwhile, cell-cycle-related pathways, including developmental growth regulation, nuclear division, and ubiquitin protein ligase activity, were significantly enriched by FIFS2. Further confirmed by gene set variation analysis (GSVA) of the hallmark gene signatures, we found FIFS1 was active for interferon, complement, inflammation, and signaling in interleukin, while FIFS2 displayed enhanced expression of the G2/M checkpoint, DNA repair, mTORC1 signaling, and targets to MYC and E2F pathways (Figure 4C). Notably, inflammatory cytokines were reported to modify the tumor microenvironment by recruiting immune cells to exert diverse immune functions [48], suggesting that proinflammatory TIME tended to be formed in FIFS1. The cell cycle is regarded as a hallmark and therapeutics target of cancer [49], which stimulates limitless tumor cell division and perturbs antigen presentation and cytokines’ secretion in FIFS2.

The complex and dynamic TIME is recognized to be important in regulating tumor growth, invasion, and metastasis [50]. We first estimated the relative fraction of stromal and immune cells in tumor tissues and found that FIFS1 had higher degrees of stromal cell (*p* = 0.024) and immune cell (*p* = 0.031) infiltration (Figure 4D). A further comprehensive evaluation of tumor purity based on a previous study [38] confirmed that FIFS2 had higher tumor purity compared with FIFS1 (*p* = 0.018, Figure 4E and Appendix A). By comparing immune cell infiltration abundance in TIME, we found the differences in immune infiltration of the two subtypes were mainly manifested in macrophages and CD8^+^ T cells. FIFS1 showed more CD8^+^ T cell infiltration (*p* = 0.029, Figure 4F and Appendix A) and more M1 polarized macrophages than FIFS2 (*p* = 3.6 × 10^−3^), while FIFS2 had more M2 polarized macrophages (*p* = 5.2 × 10^−3^, Figure 4F). By comparing the prognosis of macrophage polarized phenotypes (M1 as proinflammatory and M2 as anti-inflammatory) in the TCGA-LIHC dataset, M1 polarization (*p* = 0.02, log-rank test, Appendix A) significantly prolonged patient survival, while M2 polarization (*p* = 0.001, log-rank test, Appendix A) and a higher M2/M1 ratio (*p* = 0.001, log-rank test) were prognostic risk factors (Figure 4G).

The interaction between surface molecules and immune cells regulates the anti-tumor immune response [51], and cytokines’ production can activate both the innate and adaptive immune responses against tumor immune escape [52]. Subsequent enrichment analysis on gene sets of immune-related cell surface molecules and cytokines also confirmed the immune activation in FIFS1 (Figure 4H). In addition, we found that FIFS2 had higher stemness indices than FIFS1 (*p* = 0.008, Figure 4I), corresponding to the greater ability of tumor motile and self-renewal. The significantly lower IPS score was also observed in FIFS2, suggesting a worse immune checkpoint inhibition response and immune stimulation (*p* = 0.021, Figure 4J). Overall, we performed transcriptomics analysis to compare inter- and intra-tumor heterogeneity between two subtypes, and revealed that FIFS1 displayed stronger immune pathway activity and formed a proinflammatory tumor microenvironment via well-orchestrated reciprocal interactions between tumor cells and surrounding cells.

### 3.5. Close Radiogenomics Association between Imaging Features and Immune Response as Well as a Cell Cycle Modulating Function

We employed a network-based approach to comprehensively explore the relationship between imaging features and biological functions. First, we conducted weighted gene co-expression network analysis (WGCNA) to construct a co-expression network based on the transcriptome of the HCC patients. After determining the soft thresholding power as 8 and minimum module size as 30 (Appendix A), we identified 27 co-expression modules. By calculating the subtyping correlation and prognostic significance, eight modules were filtered and identified to be significantly associated with subtypes and had prognostic implications (Figure 5A and Appendix A). Among them, five of eight (green, darkorange, darkgreen, yellow, and red, named as PRF-related modules) were significantly correlated with radiomics features (PRF1–18, Figure 5B and Appendix A). Intriguingly, the green module was correlated with most of the PRFs (16 out of 18) and the yellow module was the only module positively correlated with FIFS2-specific PRFs (PRF11–18). To explore the potential biological functions, we performed functional enrichment analysis for PRF-related modules (Appendix A). As a result, immune-related pathways including regulation of immune cell, chemokines, cytokines, and T cell differentiation regulation were enriched by the green module (Figure 5C), while cell cycle regulation, nuclear division, and chromosome segregation were enriched by the yellow module (Figure 5D and Appendix A).

We next determined the specific biological functions correlated with radiomics features. Pathway enrichment scores of the PRF-related modules were calculated in each patient, and we then performed correlation analysis between enrichment scores and PRF eigenvalues to identify pathway–feature pairs. As a result, 5898 pathway–feature pairs were identified, and the pathways in pairs were dominated in the green module (77.92%, Appendix A and Appendix A). Consistently, except for PRF17 and PRF18, the association network based on the top five pairs of each PRF displayed predominantly immune correlations (Figure 5E), including immune cell differentiation as well as antigen-presenting and proinflammatory cytokines’ release, suggesting that radiomics features were highly correlated to anti-tumor activity such as immune response stimulation, tumor recognition, and recruitment of immune cells.

Furthermore, we identified the specific correlated genes for these PRFs. For each pathway–feature pair of the network, we calculated the correlation coefficients between the expression of genes in the pathway and the eigenvalues of the PRF. As a result, 14 genes were significantly associated with PRFs, and are hereinafter referred as PRF-related genes. The PRF-related genes were mainly distributed over the green and yellow modules and contributed to immune and cell-cycle-regulation-related functions, respectively (Table 3, Appendix A). For instance, SLAMF6 was identified to be positively correlated with PRF6 (r = 0.73, *p* < 0.001, Pearson correlation analysis) and was presented in the T helper 17 cell differentiation pathway. SLAMF6 is a member of the signaling lymphocyte activated molecule subfamily, which enhanced Th17 cell function by increasing T-cell adhesiveness through the activation of the small GTPase Rap1 [53]. LY9, another gene in that pathway, was positively correlated with PRF10 (r = 0.54, *p* < 0.001, Pearson correlation analysis), and it was reported to co-participate in IL-17 production with SLAMF6 [54] (Figure 5F). Meanwhile, CDC26 was positively correlated with FIF2-related PRF11 and was presented in the pathway of anaphase-promoting complex (APC). APC initiates the metaphase–anaphase transition by inducing the degradation of cyclin B and securin [55]. UBE2S was positively correlated with PRF12 (r = 0.66, *p* < 0.001, Pearson correlation analysis) and is involved in the regulation of the ubiquitin protein ligase activity pathway. UBE2S is a master regulator of mitosis by interacting with APC and promotes cell chemoresistance through PTEN-AKT signaling in HCC [56] (Figure 5G).

We also investigated the prognostic values of the PRF-related genes and revealed the significant prognosis or progression relevance in the HCC patients of the TCGA-LIHC dataset (Appendix A and Figure 5H). In brief, we used a network-based approach to investigate the relationship between radiomics features and biological functions, and demonstrated that texture-related features were mainly related to the regulation of immune response and cell cycle and the specific genes associated with these features had significant prognostic values in HCC patients.

### 3.6. Independent Validation for the Immunocompetent Status and Prognostic Relevance Based on the FIFS System

For further validation of the predominant radiogenomics associations between imaging features and immune-related pathways, we generated a gene-expression-based classifier based on the nearest shrunken centroid method [45] to predict the potential FIFS subtypes for HCC patients with only gene expression data. Specifically, 427 genes were selected as candidate genes by comparing the overlap of DEGs between two subtypes and the genes in PRF-related modules (green, darkorange, darkgreen, yellow, and red; Figure 6A). Based on internal 10-fold cross validation, the optimal shrinkage threshold of 1.096 was determined, and a list of 183 signature genes discriminating the FIFS subtypes was finally selected, with the lowest misclassification error of 9% (Figure 6B and Appendix A). In addition, we observed the distribution of the 183 signature genes in the modules and found that nearly half of those belong to the green module (43.1%, Appendix A). The putative biological functions of the signature genes were summarized (Appendix A) and mainly enriched in immunologic pathways. Notably, except for NFIL3, 13 out of 14 PRF-related genes overlapped with the signature genes of the classifier (Appendix A).

We then applied the classifier to two independent datasets, TCGA-LIHC and LINC-JP cohorts. As a result, patient stratification based on the predicted FIFS subtype showed distinct survival outcomes. FIFS1 had significantly better survival than FIFS2 (Figure 6C, *p* = 0.006, Figure 6D, *p* = 0.012, log-rank tests), corresponding to our prognosis observation in the training cohort. To evaluate the immunocompetent status of the stratified patients, we generated enrichment scores using the innate- and adaptive-immunity-related gene sets in a previous study [47] and performed hierarchical clustering based on the scores (Figure 6E,F). Compared with FIFS2, higher scores were observed in FIFS1, which suggested that robust immune responses against tumors were more prone to be generated in FIFS1 (Figure 6E,F). Taken together, we generated a gene-expression-based classifier to predict the potential radiomics subtypes of HCC patients, and patient stratification based on the predicted subtypes showed distinct survival outcomes and immunocompetent status, which offered the potential for inflammation-based risk stratification based on immune-related radiogenomics.

## 4. Discussion

Radiomics represents a broadly applicable framework for decoding tumor heterogeneity, and the linkage between imaging features and tumor biology functions is worthy of further exploration. In this study, we utilized integrative analysis of radiomics and genomics profiles to characterize HCC inter-tumor and intra-tumor heterogeneity. We developed a multi-view fused imaging feature subtyping model for radiomics feature fusion and patient stratification. Based on the FIFS model, two distinct radiomics subtypes that carried prognostic value were identified. Differentiated immune pathway activity and inflammatory TIME between subtypes were obtained, and the predominant radiogenomics association between texture-related and immune-related was demonstrated. These results suggested that CE-CT imaging features may aid in inflammation-based risk stratification of HCC patients.

To the best of our knowledge, the FIFS model was the first attempt to uncover putative radiomics subtypes of HCC using multi-view CE-CT feature fusion. Compared with the plain CT scan, CE-CT images generate diverse phases rather than a single one. The tumor enhancement patterns across phases and the extent of contact between the tumor and blood vessels can be highly informative in revealing the pathological diversity, location, and vascular association of the tumor [57]. Hence, we opted for CE-CT images in our study. Our FIFS model was developed based on the SNF algorithm, which is an acceptable method for radiomics subtyping by integrating multi-view radiomics features through the fusion of multiple networks [29]. Ross et al. applied the SNF method to integrate structural T1-weighted magnetic resonance imaging (MRI), single-photon emission computed tomography and clinical-behavioral assessments, and identified putative subtypes of Parkinson’s disease [58]. Han et al. uncovered potential subtypes of obsessive-compulsive disorder by integrating structural and functional MRI data [59]. In this study, we extracted tumor features in both the arterial and venous phases as the characteristics of liver dual blood supply and CE-CT multiple-phase imaging [60]. We also obtained peritumoral features as peritumoral areas tended to be altered under the influence of tumor biological aggressiveness [27]. Of note, our FIFS model distinguished two radiomics subtypes with different prognostic outcomes, although the HR value of OS and PFS was not stable enough, which was partly owing to the small sample size. In addition, the subtyping result of FIFS showed a certain relevance of the established molecular subtyping systems, suggesting that this fusion model could effectively integrate complementary multi-view imaging information and provide more clues for radiomics subtyping.

We compared the clinical differences and characterized the radiomics profiles between the FIFS subtypes. The clinical differences between subtypes were mainly AFP level and histological grade. High serum levels of AFP are associated with pathologic grade and survival across the stages of the disease [61], and histological grade is considered as a significant risk factor for HCC postoperative recurrence and is highly correlated with radiographic features [62,63]. Furthermore, by comparing radiomics profiling in two subtypes, 18 PRFs were found to be associated with prognosis, and the majority of PRFs were distributed in the tumor region of the venous phase. It was perhaps not surprising that microvascular invasion and prognostic stratification were reflected in the texture patterns of the portal venous phase [17], while tumor vascularization patterns could be reflected in the texture patterns of the arterial phase [64]. Moreover, the PRFs mainly belonged to the texture class, which reflected differences in gray level and local homogeneity. Texture features are generally considered as independent prognosis predictors in HCC patients [27,65]. Ji et al. constructed a risk prediction model to predict tumor recurrence in HCC based on 18 texture features from wavelet transforms [65], and Meng et al. established a combined radiomics clinic (CRC) model containing six texture features to predict survival in HCC patients undergoing trans-arterial chemoembolization [27]. Hence, distinct texture-dominated radiomics profiles may reflect the clinical differences in FIFS1 and FIFS2.

Previous studies have never examined the immuno-oncological characteristics of radiomics subtypes, which motivated us to elucidate the linkage between radiomics and biological function and uncover the distinct TIME status in subtypes. First, we found that pro-inflammatory pathways were significantly enriched in FIFS1. Inflammatory cytokines including TNF-α, IL-6, IFN-γ, and chemokines could exert antiproliferative and pro-apoptotic effects on tumor cells, or indirectly modulate the TIME [66,67]. The complement system is tightly functionally interlinked with inflammatory cytokines and chemokines, which co-regulated T-cell responses [68]. In turn, cell cycle and MYC pathway were activated in FIFS2. Cell cycle is regarded as a hallmark and therapeutics target of cancer, and cell cycle dysregulation in tumor cells promotes immune evasion and perturbs antigen presentation and cytokines secretion [49,69]. Moreover, MYC, one of the elevated oncogenic signaling in immunotherapy-resistant ‘cold’ tumors, has been reported to regulate the tumor microenvironment through effects on both innate and adaptive immune effector cells and immune regulatory cytokines [70]. Besides, a higher immune infiltration level was observed in FIFS1, which was mainly reflected in infiltrating immune cells of T cells and macrophages on different polarization states. The central antitumor role of T cells has been well established [71]. As previous studies have shown, classically activated (M1) macrophages and alternatively activated (M2) macrophages were in opposite polarization states on tumor-associated macrophages. M1 macrophages exerted inflammatory and anti-tumorigenic effects through upregulated immune responses and immune surveillance on the secretion of cytokines and chemokines, while M2 macrophages exerted pro-tumorigenic, immunosuppressive effects [72,73]. Furthermore, higher immune-related cell surface molecules, cytokines, and more active immunogenicity status were observed in FIFS1, which can lead to the conclusion that FIFS1 harbors more active TIME with the characteristic of ‘hot’ tumors and is more susceptible to immunotherapy [74]. Taken together, subtypes identified by the FIFS model showed a clear distinction in immuno-oncological characteristics.

Moreover, we used integrated network analysis to investigate the potential radiogenomics linkages and uncovered the predominant association between texture-related features and immune-related pathways. For instance, SLAMF6 gene was identified to be positively correlated with PRF6 and was presented in the T Helper 17 Cell Differentiation pathway. SLAMF6 is a member of the signaling-lymphocyte-activated molecule subfamily, which enhanced Th17 cell function by increasing T-cell adhesiveness through activation of the small GTPase Rap1 [53]. LY9, another gene in that pathway, was positively correlated with PRF10 and has been reported to co-participate in IL-17 production with SLAMF6 [54]. Consistently, previous studies have also supported the linkage between immuno-oncological characteristics and texture features in HCC. Hectors et al. found that texture features could reflect immune status in HCC patients, and were highly correlated with mRNA and protein expression of PDL1 and the markers of macrophages (CD68) and T cells (CD3) [23]. Su et al. developed a radiomics model for tumor-infiltrating lymphocyte status based on GLCM and GLSZM texture features [40]. Ming et al. also revealed that cell cycle pathway exhibited significant associations with SurfaceVolumeRatio (PRF9 and PRF10 in our study) [75,76]. Hence, these results demonstrated the credibility of links between immuno-oncological characteristics and imaging features. Furthermore, we applied external datasets to validate the radiogenomics association. The gene-expression-based classifier stratified patients into two validation cohorts with distinct prognostic outcomes and immunocompetent status, suggesting that the immunodominance ‘hot’ tumors phenotype could be identified based on radiogenomics linkages in the FIFS model. Accordingly, radiogenomics associations provided important information regarding prognosis status and immuno-oncological characteristics.

Of course, there were still some limitations in this study. First, the relatively small sample size and non-readily available radiogenomics HCC datasets limited our ability to detail the characterization of the FIFS subtypes and rigorous validation of radiogenomics association. To verify our conclusion as much as possible, we conducted comparisons between the FIFS system and molecular subtyping systems with large sample sizes to verify the credibility of our subtyping results. Besides, we conducted leave-one-out cross-validation using the random forest classifier based on 18 PRFs, with an accuracy of 92.9% in our discovery cohort (Appendix A), and trained a gene-expression-based classifier to validate the radiogenomics linkage. Future studies of HCC radiogenomics with larger sample sizes may be able to more definitively address this shortcoming. Second, images obtained with CT scanners of different manufacturers in TCIA may result in a bias in generalizability. We performed a batch effect correction during data pre-processing in order to reduce the deviation. Finally, the preliminary radiogenomics association needs to be further validated in more multicenter prospective cohorts. Considering that radiomics has unique non-invasive advantages for dissecting tumor heterogeneity, we also hope that comprehensive consensus HCC subtypes could take radiomics features into account; after all, imaging examination is still one of the most common methods for HCC, as is the serologic test.

## 5. Conclusions

In conclusion, we developed a multi-view fused imaging feature subtyping model for dissecting immuno-oncological characteristics and prognoses of HCC from a non-invasive perspective. We identified two radiomics subtypes with clinical relevance and harbored distinct texture-dominated radiomics profiles, coupled with differential inflammatory pathway activity and TIME status. The predominant radiogenomics association between texture-related features and immune-related pathways was elucidated and followed by independent validation. Our study delineated the potential associations between radiomics features and immuno-oncological characteristics, and CE-CT multi-view features may serve as non-invasive predictors of inflammation-based risk stratification among HCC patients. We hoped that our study could provide the theoretical rationale and feasibility for precision treatments of HCC in the future.

## Figures and Tables

**Figure 1 cancers-15-02338-f001:**
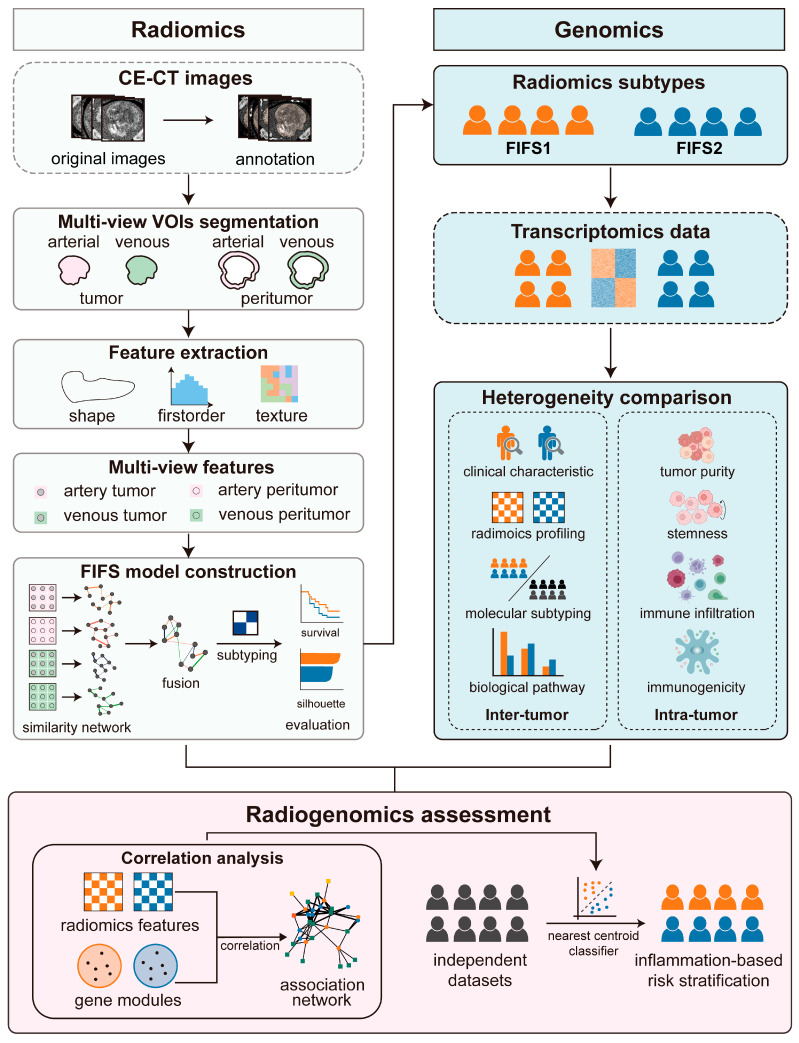
Schematic diagram of the study. We divide the workflow into three main steps. First, we segment CE-CT images to acquire multi-view VOIs (tumor and peritumoral regions in the arterial and venous phase, respectively), followed by extracting multi-view imaging features. Next, the FIFS model is developed for feature fusion and radiomics subtype identification. Based on the corresponding gene expression profiles and imaging features, we compare inter- and intra-tumor heterogeneity between subtypes. Finally, radiogenomics association is demonstrated by integrating feature–pathway network analysis and validated by inflammation-based risk stratification in two independent cohorts.

**Figure 2 cancers-15-02338-f002:**
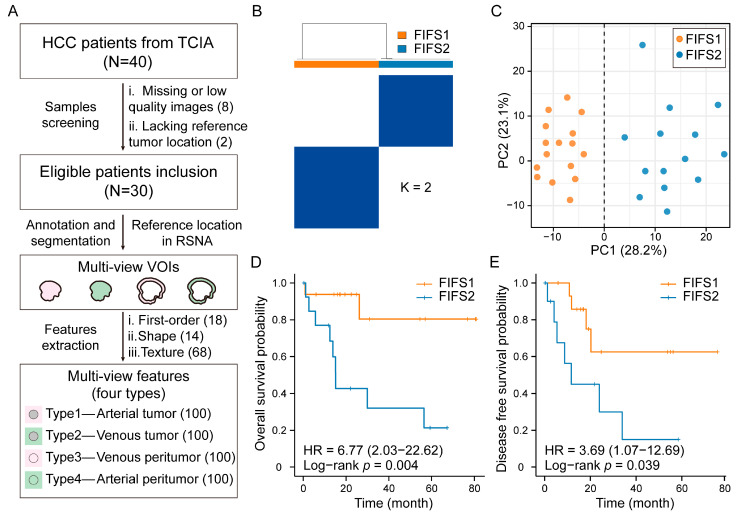
HCC patients stratified into two subtypes by multi-view radiomics features. (**A**) The process of multi-view feature extraction. After patient screening and multi-view VOIs’ segmentation, multi-view features are obtained from four types of VOIs. (**B**) Consensus clustering of HCC patients based on CP1 fused features reveals two distinct subtypes. The strength of the blue color is proportional to the frequency at which samples have been clustered together. (**C**) Principal component analysis of HCC patients based on CP1 fused features separates FIFS1 and FIFS2. (**D**,**E**) Kaplan–Meier plots show the prognosis association of subtypes with overall survival (**D**) and disease-free survival (**E**) outcomes.

**Figure 3 cancers-15-02338-f003:**
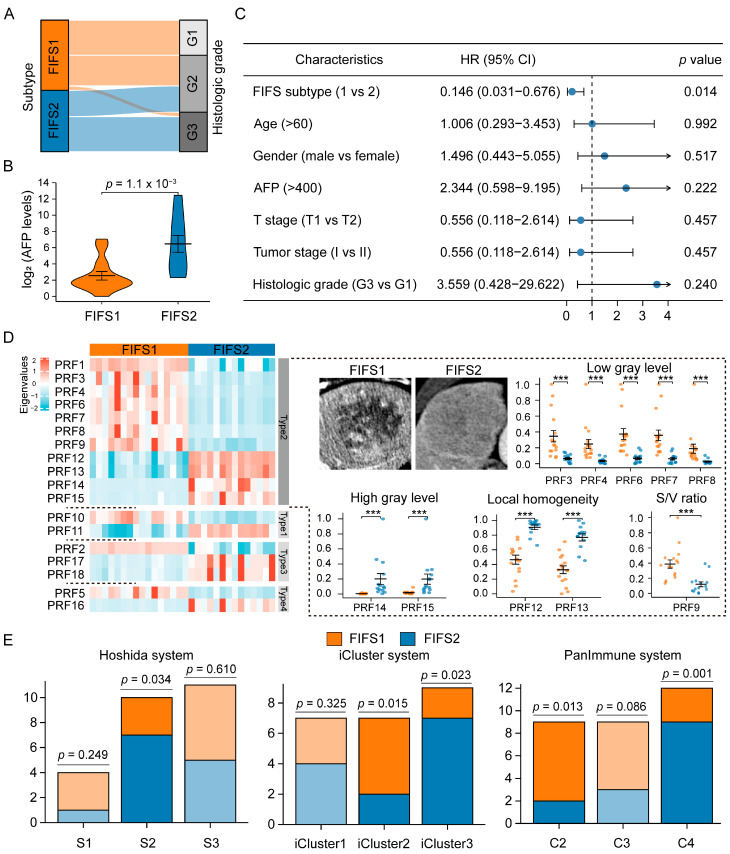
Clinical and radiomics characterization of radiomics-based subtypes. (**A**) The alluvial diagram shows the association between radiomics subtypes and histologic grades. (**B**) The violin plot outlines serum AFP values in FIFS1 and FIFS2, respectively. (**C**) The result of Cox regression analysis for the FIFS system and clinical risk factors. (**D**) Heatmap of the eigenvalues’ distribution of PRFs in FIFS1 and FIFS2. The rows are split by imaging feature types according to phase and region information. The representative images in the tumor region of the venous phase (Type 2) of FIFS1 and FIFS2 are presented, and grouped dot plots of PRF eigenvalues are indicated to the right of the heatmap. The eigenvalues of each PRF are row-normalized. ***, *p* < 0.001. (**E**) The distribution of three HCC molecular systems in FIFS subtypes. The bar in solid color is statistically significant and the bar in transparent color represents no statistically significant trend. The statistical significance is estimated by hypergeometric tests. S/V ratio, surface–volume ratio.

**Figure 4 cancers-15-02338-f004:**
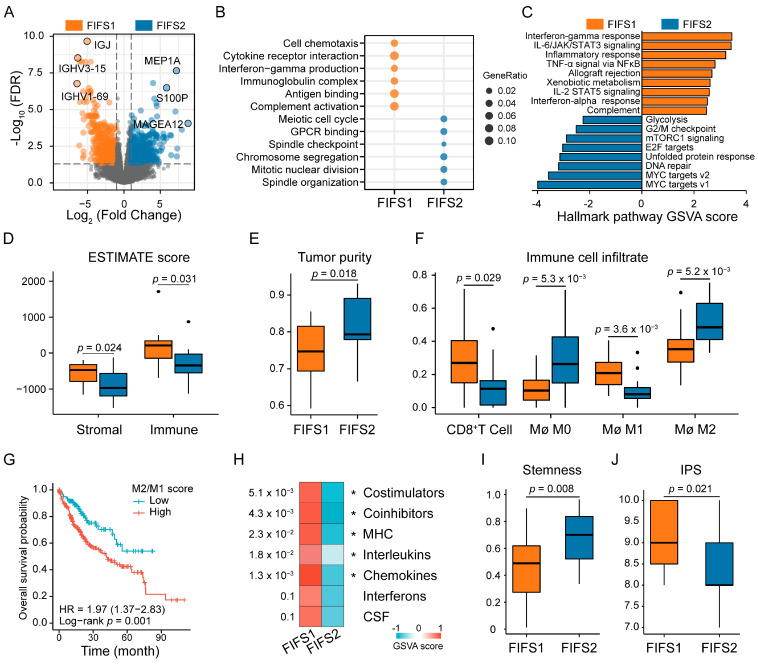
Radiomics subtypes reveal distinct tumor biological functions and TIME status. (**A**) The volcano plot shows DEGs between subtypes. The top three significant DEGs are labeled. (**B**) The dot plot represents the enriched gene sets by GO and KEGG enrichment analysis. (**C**) Differences in hallmark signatures activities scored by GSVA between FIFS1 and FIFS2. The orange color indicates significantly activated pathways in FIFS1, and the steel blue color indicates significantly activated pathways in FIFS2. (**D**) Higher stromal scores and immune scores are observed in FIFS1. Dots represent outliers. (**E**) Tumor purity score across subtypes inferred by the CPE algorithms. (**F**) Distribution of immune cell infiltration between two subtypes. Dots represent outliers. (**G**) Kaplan–Meier plot of overall survival rates in TCGA-LIHC samples stratified by M2 macrophages/M1 macrophages ratio. (**H**) The heatmap shows higher overall expression levels (GSVA scores) of immune-related cell surface molecules (costimulators, coinhibitors, and major histocompatibility complex) and cytokines (interleukins, chemokines, interferons, and colony-stimulating factors) in FIFS1. *, *p* < 0.05. (**I**,**J**) Boxplots show the stemness index (**I**) and IPS scores (**J**) in FIFS1 and FIFS2. Mø, Macrophages.

**Figure 5 cancers-15-02338-f005:**
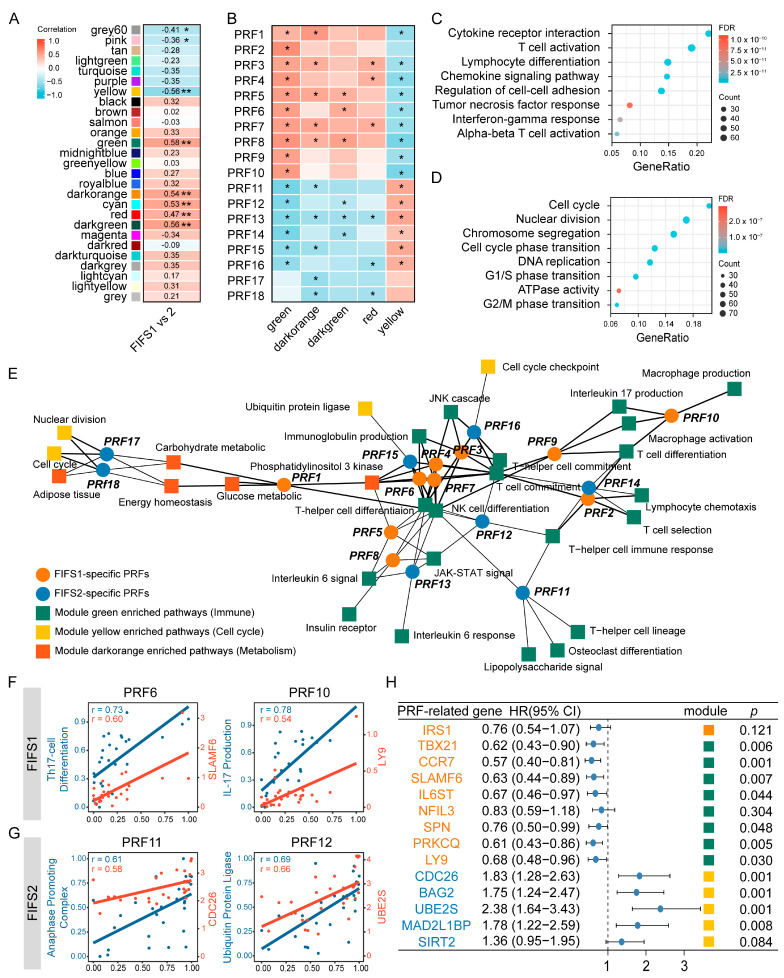
Radiomics features are associated with immune response and cell cycle modulating function. (**A**) The correlation matrix between co-expression modules and FIFS subtyping phenotypes. Eight modules are significantly associated with FIFS phenotypes and harbored prognostic values are indicated with an asterisk. *, *p* < 0.05; **, *p* < 0.01. (**B**) Correlations between PRFs and the PRF-related modules are displayed as a heatmap. *, *p* < 0.05. (**C**,**D**) Enrichment analysis results of genes within the green module (**C**) and the yellow module (**D**). (**E**) An association network of the top five pathway–feature pairs for each PRF. The circles in orange and steelblue indicate FIF1-specific and FIF2-specific PRFs, respectively. The squares in green, yellow, and orange indicate the specific biological functions of corresponding color-coded modules. (**F**,**G**) The PRFs are significantly correlated with biological functions in FIFS1 (**F**) and FIFS2 (**G**). The points in blue indicate the correlation between the most relevant biological pathway and PRFs and the points in red indicate the correlation between the PRF-related genes and PRFs. The correlation coefficients are indicated in the upper left corner of the graph. (**H**) The forest plot shows the Cox regression result of the PRF-related genes. The genes in orange color indicate the FIFS1-specific PRF-related genes and the genes in steelblue color are FIFS2-specific. Squares are color-coded based on corresponding module color information in (**A**) and indicate the modules to which PRF-related genes belong.

**Figure 6 cancers-15-02338-f006:**
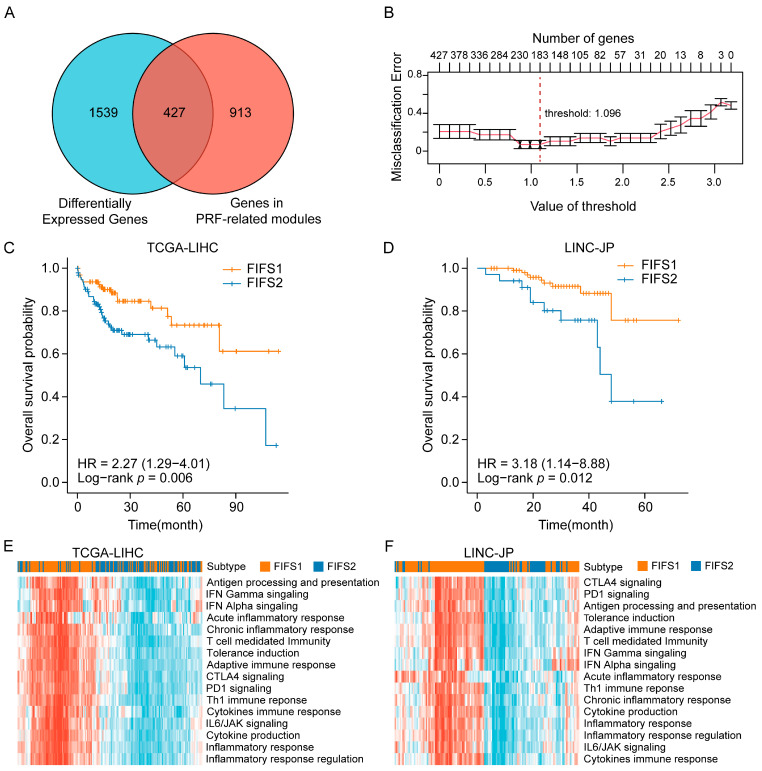
Independent validation for the immunocompetent status and prognostic relevance based on the FIFS system. (**A**) Venn diagram shows the overlapping genes between DEGs and the genes of PRF-related modules. (**B**) The line chart shows the number of candidate genes and misclassification errors at different thresholds. (**C**,**D**) Kaplan–Meier survival plots show the overall survival of HCC patients stratified by classifier prediction in the TCGA-LIHC (**C**) and LINC-JP datasets (**D**). (**E**,**F**) Heatmaps show the innate- and adaptive-immunity-related pathway enrichment scores for patients of FIFS1 and FIFS2 in TCGA-LIHC cohort (**E**) and the LINC-JP cohort (**F**). Rows show immune-related pathway enrichment scores (z-scores) and columns show the patients stratified by the classifier.

**Table 1 cancers-15-02338-t001:** Baseline patient characteristics.

Characteristics	Discovery Cohort (*n* = 30)	Validation Cohort 1 TCGA-LIHC (*n* = 192)	Validation Cohort 2 LINC-JP (*n* = 142)	*p*
Age (year)	66 (56, 68)	59.5 (51, 69)	69 (62, 75)	<0.001
Gender				0.649
male	20	139	97	
female	10	53	45	
AFP (ng/mL)	73.5 ± 168.2	186.6 ± 496.6	NA	0.765
T stage				0.920
T1	23	148	NA	
T2	7	43	NA	
Tumor stage				<0.001
stage I	23	149	36	
stage II	7	43	106	
Histology grade				0.006
G1	6	21	17	
G2	14	93	83	
G3	10	65	31	
G4	0	11	0	
Treatment methods				<0.001
segmentectomy	19	103	2	
lobectomy	8	70	0	
extended lobectomy	3	7	0	
total hepatectomy with transplant	0	1	0	
TACE	0	0	25	
chemotherapy	0	0	1	
Follow-up duration (day)	552.0 (383.5, 1459.0)	631.5 (381.2, 1289.0)	870.0 (570.5, 1132.5)	0.899

**Table 2 cancers-15-02338-t002:** Characteristics of the subtype-specific prognostic radiomics features.

Feature	Feature Name	Subtype	Type	Phase	Region	Class	HR (95% CI)	*p*
PRF1	Minimum	FIFS1	Type2	venous	tumor	first-order	0.17 (0.05–0.55)	0.008
PRF2	LargeAreaHighGrayLevelEmphasis	FIFS1	Type3	venous	margin	texture-glszm	0.07 (0.02–0.24)	0.001
PRF3	LargeDependenceLowGrayLevelEmphasis	FIFS1	Type2	venous	tumor	texture-gldm	0.28 (0.08–0.92)	0.043
PRF4	LowGrayLevelZoneEmphasis	FIFS1	Type2	venous	tumor	texture-glszm	0.19 (0.06–0.61)	0.016
PRF5	ShortRunHighGrayLevelEmphasis	FIFS1	Type4	artery	peritumor	texture-glrlm	0.09 (0.03–0.28)	0.003
PRF6	LowGrayLevelEmphasis	FIFS1	Type2	venous	tumor	texture-gldm	0.20 (0.06–0.66)	0.021
PRF7	LowGrayLevelRunEmphasis	FIFS1	Type2	venous	tumor	texture-glrlm	0.19 (0.06–0.61)	0.016
PRF8	SmallAreaLowGrayLevelEmphasis	FIFS1	Type2	venous	tumor	texture-glszm	0.23 (0.07–0.76)	0.039
PRF9	SurfaceVolumeRatio	FIFS1	Type2	venous	tumor	shape	0.06 (0.02–0.22)	<0.001
PRF10	SurfaceVolumeRatio	FIFS1	Type1	artery	tumor	shape	0.06 (0.02–0.21)	<0.001
PRF11	Idmn	FIFS2	Type1	artery	tumor	texture-glcm	3.74 (1.13–12.38)	0.035
PRF12	Idmn	FIFS2	Type2	venous	tumor	texture-glcm	14.84 (4.45–49.53)	0.001
PRF13	Idn	FIFS2	Type2	venous	tumor	texture-glcm	5.34 (1.63–17.50)	0.016
PRF14	LargeAreaHighGrayLevelEmphasis	FIFS2	Type2	venous	tumor	texture-glszm	15.61 (4.65–52.40)	<0.001
PRF15	LargeDependenceHighGrayLevelEmphasis	FIFS2	Type2	venous	tumor	texture-gldm	6.28 (1.89–20.82)	0.007
PRF16	LongRunLowGrayLevelEmphasis	FIFS2	Type4	artery	peritumor	texture-glrlm	4.88 (1.49–15.92)	0.024
PRF17	DifferenceVariance	FIFS2	Type3	venous	peritumor	texture-glcm	4.73 (1.45–15.41)	0.028
PRF18	Contrast	FIFS2	Type3	venous	peritumor	texture-glcm	5.34 (1.63–17.48)	0.016

**Table 3 cancers-15-02338-t003:** Characteristics of the PRF-related genes.

PRF-Related Gene	Imaging Feature	Subtype Specific	Pathway	Module	Correlation Coefficient	*p*
IRS1	PRF1	FIFS1	Positive Regulation of Cellular Carbohydrate Metabolic Process	Darkorange	0.534	0.003
TBX21	PRF2	FIFS1	T Cell Differentiation Involved in Immune Response	Green	0.592	0.001
CCR7	PRF3	FIFS1	Regulation of JNK Cascade	Green	0.679	<0.001
SLAMF6	PRF4	FIFS1	CD4 Positive or CD8 Positive Alpha Beta T Cell Lineage Commitment	Green	0.611	0.001
IL6ST	PRF5	FIFS1	JAK-SKAT Signaling Pathway	Green	0.556	0.002
SLAMF6	PRF6	FIFS1	T Helper 17 Cell Differentiation	Green	0.595	0.001
NFIL3	PRF7	FIFS1	Natural Killer Cell Differentiation	Green	0.577	0.001
SPN	PRF8	FIFS1	CD4 Positive or CD8 Positive Alpha Beta T Cell Lineage Commitment	Green	0.518	0.004
PRKCQ	PRF9	FIFS1	Positive Regulation of Interleukin 17 Production	Green	0.581	0.001
LY9	PRF10	FIFS1	Positive Regulation of Interleukin 17 Production	Green	0.54	0.003
CDC26	PRF11	FIFS2	Anaphase Promoting Complex	Yellow	0.578	0.001
UBE2S	PRF12	FIFS2	Regulation of Ubiquitin Protein Ligase Activity	Yellow	0.662	<0.001
BAG2	PRF13	FIFS2	Regulation of Ubiquitin Protein Ligase Activity	Yellow	0.601	0.001
UBE2S	PRF14	FIFS2	Positive Regulation of Ubiquitin Protein Transferase Activity	Yellow	0.527	0.004
UBE2S	PRF15	FIFS2	Regulation of Ubiquitin Protein Ligase Activity	Yellow	0.623	<0.001
MAD2L1BP	PRF16	FIFS2	Regulation of Mitotic Cell Cycle Spindle Assembly Checkpoint	Yellow	0.539	0.003
SIRT2	PRF17	FIFS2	Positive Regulation of Meiotic Cell Cycle	Yellow	0.486	0.008
SIRT2	PRF18	FIFS2	Regulation of Meiotic Nuclear Division	Yellow	0.584	0.001

## Data Availability

The CE-CT images of HCC patients were collected from The Cancer Imaging Archive. The corresponding transcriptomics data were downloaded from The Cancer Genome Atlas. Two additional cohorts were collected from The Cancer Genome Atlas and International Cancer Genome Consortium.

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
