# Peer review of "Multi-View Radiomics Feature Fusion Reveals Distinct Immuno-Oncological Characteristics and Clinical Prognoses in Hepatocellular Carcinoma"

_cancers, 2023, doi:10.3390/cancers15082338_

Round 1

Reviewer 1 Report

The authors have done an excellent job to present all their findings together in a coherent manner. A lot of effort has been given to produce this work. Best of luck to all of them.

Below are the comments:

Short summary:

-        Please remove the summary and merge some of that information with the abstract.

Introduction:

-        Please briefly mention what is  intra and inter-tumor heterogeneity. Many interested readers maybe unaware of this term.

-        The authors mentioned: “Due to immense challenges in deciphering therapeutic strategies, characterizing the landscape of HCC is paramount.” Please mention some of the challenges.

-        Can the authors clarify why they opted for CE-CT images? Why not use CT and PET images?

-        The authors mentioned: “A previous systematic review of 54 included HCC studies has acknowledged the predictive value of radiomics features in the microvascular invasion, prognosis and early recurrence.” Please mention some of them, to grow more interest as the reader continues to radiogenomics.

-        In general, the motivation is explained well, and the contribution is stated clearly.

Materials and methods:

-        Please provide full form of LIHC, LINC-JP, and FIFS.

-        The methods are well explained.

-        Please clarify a bit more on the FIFS1 & FIFS2 subtypes. These are the two subtype categories in study. FIFS1 and FIFS2 creates confusion by showing two FIFS models used.

-        A lot of information is given and it is difficult to follow-up. Please give another diagram to illustrate how the study of intra and inter-tumor heterogeneity has been conducted.

Results:

-        In Figure 2D & 2E, the FIFS2 shows similar overall and disease-free survival probabilities. Can the authors please explain why so in few lines.

-        In line 310, the authors mentioned that “we found that the FIFS model was an independent prognostic factor”. This is a very important finding. Please mention it in the abstract.

-        A summary of all the findings (in few lines) of each section can be added at the end of the results section to comprehend all the information and findings in a nut-shell. This should serve a s a takeaway message to the reader.

References:

-        Some of the articles cited are more than 10 years old. Please replace them with more recent ones.

Reviewer 2 Report

The study developed a multi-view CE-CT radiomics features fusion model for dissecting immuno-oncological characteristics and prognoses of HCC from a non-invasive perspective. The study identified two radiomics subtypes with clinical relevance and harbored distinct texture-dominated radiomics profiles, as well as coupled with differential inflammatory pathway activity and TIME status. The study delineated the potential associations between radiomics features and immuno-oncological characteristics, and CE-CT multi-view features may serve as non-invasive predictors of inflammation-based risk stratification among HCC patients. The study's approach represents a novel way to integrate imaging and genomic data to better understand HCC biology and improve patient outcomes. Further validation in multicenter prospective cohorts is needed to fully realize the potential of this approach in clinical practice.

There are some comments:

1.     Page 2, line 78: “We developed a multi-view CE-CT radiomics features fusion model, and identified two radiomics subtypes with distinct biological functions and inflammatory TIME status. Texture-related imaging features showed pre-dominant radiogenomics associations with immune-related biological functions, and were subsequently verified in independent cohorts. Taken together, our study revealed significant linkages between multi-view CE-CT imaging features and immuno-oncological characteristics in HCC, and these findings may provide the theoretical rationale and feasibility for non-invasive inflammation-based risk stratification.” The paragraph summarizes the results and should move to the Discussion, not in the introduction.

2.     Please give a table to show the clinical information of the original cohort and two validation cohorts shown in Figure 3, including age, gender, AFP, T stage, tumor stages, histology grade, treatment methods, and follow-up duration. Please give P values to show the differences between cohorts. Did all patients receive surgical resection to get tumor specimens?

3.     Page 3, line 94: “Besides, two additional cohorts (192 HCC patients without images in TCGA-LIHC, and 142 HCC patients in LINC-JP) were employed for further radiogenomics association validation analysis.” Please give the reference or website address.

4.     The Multiview features of the four types in Table 1 and Figure 2A are not consistent. Please check it.

5.     Line 589: “In this study, we extracted tumor features in both the arterial and venous phases since the characteristics of liver dual blood supply and CE-CT multiple-590 phase imaging[52]. We also obtained peritumoral features since peritumoral areas tended to be altered under the influence of tumor biological aggressiveness, microinvasion, and metastasis[53]”. References 52 and 53 did not match the statements. Please revise it.
